# Biochemical properties and biological potential of *Syzygium heyneanum* with antiparkinson's activity in paraquat induced rodent model

Malik Saadullah[1], Hafsa Tariq[1], Zunera Chauhdary [2]*, Uzma Saleem[2], Shazia Anwer Bukhari[3]*, Amna Sehar[1], Muhammad Asif[4], Aisha Sethi[5]

1 Department of Pharmaceutical Chemistry, Government College University Faisalabad, Faisalabad, Pakistan, 2 Department of Pharmacology, Government College University Faisalabad, Faisalabad, Pakistan, 3 Department of Biochemistry, Government College University Faisalabad, Faisalabad, Pakistan, 4 Department of Pharmacology, Islamia University Bahawalpur, Bahawalpur, Pakistan, 5 Department of Pharmaceutics, Government College University Faisalabad, Faisalabad, Pakistan

* zunerach@yahoo.com (ZC); shaziabukhari@gcuf.edu.pk (SAB)

**Data Availability Statement:** All relevant data are within the manuscript.

## Abstract

*Syzygium heyneanum* is a valuable source of flavonoids and phenols, known for their antioxidant and neuroprotective properties. This research aimed to explore the potential of *Syzygium heyneanum* ethanol extract (SHE) in countering Parkinson's disease. The presence of phenols and flavonoids results in SHE displaying an $IC_{50}$ value of 42.13 when assessed in the DPPH scavenging assay. Rats' vital organs (lungs, heart, spleen, liver, and kidney) histopathology reveals little or almost no harmful effect. The study hypothesized that SHE possesses antioxidants that could mitigate Parkinson's symptoms by influencing α-synuclein, acetylcholinesterase (AChE), TNF-α, and IL-1β. Both *in silico* and *in vivo* investigations were conducted. The Parkinson's rat model was established using paraquat (1 mg/kg, i.p.), with rats divided into control, disease control, standard, and SHE-treated groups (150, 300, and 600 mg/kg) for 21 days. According to the ELISA statistics, the SHE treated group had lowers levels of IL-6 and TNF-α than the disease control group, which is a sign of neuroprotection. Behavioral and biochemical assessments were performed, alongside mRNA expression analyses using RT-PCR to assess SHE's impact on α-synuclein, AChE, TNF-α, and interleukins in brain homogenates. Behavioral observations demonstrated dose-dependent improvements in rats treated with SHE (600 > 300 > 150 mg/kg). Antioxidant enzyme levels (catalase, superoxide dismutase, glutathione) were significantly restored, particularly at a high dose, with notable reduction in malondialdehyde. The high dose of SHE notably lowered acetylcholinesterase levels. qRT-PCR results indicated reduced mRNA expression of IL-1β, α-synuclein, TNF-α, and AChE in SHE-treated groups compared to disease controls, suggesting neuroprotection. In conclusion, this study highlights *Syzygium heyneanum* potential to alleviate Parkinson's disease symptoms through its antioxidant and modulatory effects on relevant biomarkers.

**Funding:** The author(s) received no specific funding for this work.

**Competing interests:** The authors have declared that no competing interests exist.

**Abbreviations:** PD, Parkinson's disease; SNpc, substantia nigra pars compacta; DPPH, 2, 2-diphenyl-1-picrylhydrazyl; HPLC, High-performance liquid chromatography; TBA, Thiobarbituric acid; MDA, malondialdehyde; CAT, catalase; SOD, Superoxide dismutase; TCA, trichloroacetic acid; AchE, Acetyl Cholinesterase; SHE, *Syzygium heyneanum* ethanolic extract; CV, central vein; PV, portal vein; BD, bile duct; HA, hepatic artery; BR, bronchiole; AD, alveolar duct; MF, myocardial fibrils; CN, central nuclei; US, urinary space; DCT, distal convoluted tubules; PCT, proximal convoluted tubules; WP, white pulp; RP, red pulp.

## 1. Introduction

Parkinson's disease (PD) is diagnosable clinical illness containing a variety of etiologies and clinical manifestations [1]. With the exception of an infectious origin, the frequency of Parkinson's disease, a neurological condition, is rising quickly worldwide [2]. PD is becoming increasingly popular among the old age people, its symptoms are memory loss, difficulties in movements and sleep. Parkinson's disease(PD) is complicated neurological condition, with deceasing dopaminergic neurons in the substantia nigra pars compacta (SNpc) that creates deficiency of dopamine [3] which causes unintended movements. Mitochondrial dysfunction is also a major cause of neurodegenerative diseases specially PD [4]. Proteostasis changes, oxidative stress, and neuroinflammation are generally acknowledged as important components contributing to the pathology of PD, despite the fact that its aetiology is still largely unknown [5, 6]. Levodopa and carbidopa-containing drugs including Sinemet, Paracopa, Rytary, and Duopa are widely used as recognized treatments for the motor symptoms of PD [7], but there chronic administration causes hypertension, ulcer, depression and toxicity [8]. Consequently a potent complementary herbal medicinal agent is a vital clinical requirement in the modern world [9].

PD may occasionally be inherited being by cause of particular genetic alterations [10]. The SNCA, LRRK2, and PARK2 genes were all identified to enhance the probability of developing Parkinsonism [11]. The danger of developing PD has been linked to exposure to several chemicals, including pesticides and herbicides [12]. Head injuries and other environmental elements may potentially raise the possibility of developing PD [13, 14]. Age affects the probability of getting PD, and people over 60 are the most frequently identified cases of the disease[15, 16]. Oxidative stress, which develops when the body's levels of free radicals and antioxidants are out of equilibrium, may harm neurons in the brain and result in PD [17, 18]. The onset of PD may be influenced by brain inflammation [19, 20]. Mitochondria are the energy-producing structures within cells, and dysfunction of these structures may contribute to the evolution of Parkinsonism [21–23].

*Syzygium heyneanum* has a place with family Myrtaceae which is the group of dicotyledonous plants [24]. Myrtaceae contains roughly 5,950 species in around 132 genera. It is widely expressed in warm, tranquil tropical regions of the world. All species have bloom portions in products four and five, are woody, and contain fundamental oils [25]. *Syzygium heyneanum* Powdered bark is consumed orally with water to treat indigestion and stomach problems [26]. Powder of *Syzygium heyneanum* stem bark is mixed with lime and butter milk and is given orally to cure dysentery [27, 28]. To treat a wound, bark paste of *Syzygium heyneanum* is applied topically to the damaged area of the skin [29]. Plant's eastern or western side must be used to harvest the bark. Stem bark of this plant is also used for the treatment of epilepsy [30]. *Syzygium heyneanum* stem bark is boiled with water, and this water is used as a preventive measure from sunstroke [31].

There are numerous secondary metabolites found in the Syzygium genus, including terpenoids, chalcones, flavonoids, lignans, alkyl phloroglucinols, hydrolysable tannins, and chromone derivatives [32]. These compounds have bioactivities including antidiabetic, antifungal, anticancer, anti-inflammatory, antibacterial, antimicrobial, antioxidant, cytotoxic, anti-HIV, antidiarrheal [32]. *S. heyneanum* phytochemicals was not reported with their medicinal use against neurodegenerative disorders. Therefore, the present research was intended to analyze phytochemical investigation, qualitative analysis of chemical constituents, toxicity study and evaluation of anti-Parkinson's activity of *S. heyneanum* against paraquat induced rat model through behavioral and biochemical analysis.

## 2. Material and methods

### 2.1 Plant collection and extraction

In July 2022, *S. heyneanum* leaves were taken from the Botanical garden of Punjab University in Lahore. The Government College University in Faisalabad, Pakistan, verified the leaves, and a voucher was filed in herbarium NO: 423/GCUF. The leaves were washed to remove dirt and unnecessary items before being dried for 15 days in the shade. After that, leaves were pounded into a fine powder and macerated 250 g of this powder for 24 hours in a tightly covered glass container with 350 ml of ethanol [33]. Macerated plant was filtered by whatman filter paper. This procedure was repeated 3 times to get maximum extract. Extract was covered with pitted aluminum foil for evaporation. After 20 days partially solid extract was obtained which was retained at room temperature in closed glass container [33].

### 2.2 Estimation of secondary metabolites from plant extract

**2.2.1 Total flavonoid content test.** To determine the total flavonoid contents, a dosage response curve of standard and a linear regression equation were plotted using quercetin as the standard. A test tube containing 4.6 mL of distilled water, along with 0.1 mL of 1M potassium acetate, 0.2 mL of standard and sample concentrations, and 0.1 mL of 10% aluminum nitrate, was placed in an incubator at room temperature for duration of 50 minutes. To create the blank solution, all the reagents were combined, with the exception of the sample or standard. At 415 nm, absorbance was observed. The amount of quercetin equaling mg of total Flavonoid content was determined [34]. Three times this experiment was repeated. For estimating the total flavonoids, the formula shown below was utilized [33].

$$\text{Total flavonoids} = \frac{\text{Quercetin equivalent} \left( \mu \frac{\text{g}}{\text{ml}} \right) \times \text{extract volume}}{\text{Sample (g)}}$$

**2.2.2 Total polyphenolic content test.** Using Saleem et al. approach, the total polyphenolic contents were determined. Various gallic acid concentration ranges were utilized to plot the standard curve. In test tubes, 0.2 ml of standard as well as sample concentrations were held separately, combined along 0.2 mL of the phenol reagent from Folin-Ciocalteu. Each test tube received 1 mL of 15% sodium carbonate solution after 4 minutes of room temperature incubation. After mixing, each test tube was incubated for 2 hours at 25˚C. At 760 nm, the absorption was detected. All reagents were combined in the composition of a blank, save from the sample. The gallic acid standard curve was utilized to convert the entire amount of polyphenols to mg of gallic acid equivalent in a linear regression equation [35]. The process was carried out in triplicate, and the following equation was utilized to measure the approximate total amount of polyphenols in the sample [33].

$$\text{Total polyphenolics} = \frac{\text{Gallic acid equivalent} \left( \mu \text{ g mL} \right) \times \text{extract volume}}{\text{sample (g)}}$$

### 2.3 In vitro antioxidant assay

**2.3.1 DPPH scavenging assay.** According to Saleem et al. the 2, 2-diphenyl-1-picrylhydrazyl (DPPH) radical scavenging assay was used to evaluate the sample's antioxidant potential [36]. This method calls for mixing 3mL of sample with 1mL of freshly made 0.004% DPPH in methanol solution, which was then left in the dark for 30 minutes. Then, at 517 nm, absorption was discovered. A reaction combination with a low absorbance has a high level of radical scavenging activity. Analyses were also done on the typical antioxidant activity of ascorbic acid

[37]. The solution devoid of plant extract served as the control. Three duplicates of each test were run. Following is a formula for calculating the % inhibition of DPPH radical samples.

$$DPPH\ Inhibition(\%) = \frac{Blank\ absorbance\ (A_0) - Sample\ absorbance\ (A_1)^* 100}{Blank\ absorbance\ (A_0)}$$

Where

$A_1$ = Absorbance of sample.

$A_0$ = Absorbance of blank.

The plot depicting the percentage of inhibition in relation to the concentration of the sample was employed to determine the sample concentration at which 50% inhibition occurs. The experiments were conducted three times to ensure accuracy. Ascorbic acid was utilized as the positive control in the experimentation process.

## 2.4 HPLC analysis

The substance underwent filtration utilizing a 0.45 μm Millex-HV PVDF membrane, manufactured by Millipore situated in New Bedford, Massachusetts. High-performance liquid chromatography (HPLC) analysis was conducted using a Shimadzu chromatograph that featured a ternary pump (Shimadzu LC-20AT) and a diode array detector (Shimadzu SPD-M20A) from Kyoto, Japan. The analysis also employed an analytical column (Phenomenex® ODS 100 A, dimensions 250 mm x 4.60 mm, particle size 5 μm) along with a C18 guard column (dimensions 2.0 cm x 4.0 mm; particle size 5 μm). The analysis was facilitated using LC Solutions software (version 1.25) [38].

A gradient chromatography technique was employed, employing a mobile phase comprising acetonitrile and water, which flowed at a rate of 1 ml/min. The column temperature was maintained at 25°C, and each injection involved a volume of 20 μl. The final acetonitrile/water ratio was 8:2, which was an improvement over the initial 2:8 v/v ratio. The mobile phase was ready for use every day and sonicated to remove gas. 450 to 200 nm were used to observe the UV spectra. The ethanol used to create all of the standard solutions and extracts. The concentrations of the reference solutions (12.5, 25.0, 100.0, 150.0, 250.0, and 500.0 g/mL) were employed, whereas the concentration of the solution chemical was 2000 g/mL.

## 2.5 Approval from animal ethics committee

The National Research Council's Commission on Life Sciences University, Institute of Laboratory Animal Resources, 1996 regulations was followed in the animal research that was done. The Institutional Review Board at GC University gave its blessing to this study, which was conducted under the reference number GCUF/ERC/265.

## 2.6 Toxicity studies

**2.6.1 Animal husbandry.** Wistar rats weighing (250–300 g), aged 10–12 weeks were taken. Rats were obtained from animal house of Government College University Faisalabad. For acclimatization, the rats were retained in animal house under standard environmental conditions i.e. in 12 hrs. a day and 12 hrs. night cycle, temperature 24±2°C, relative humidity 30–60% for five days before starting dosing in Government College University Faisalabad. The rats were fed with water ad libitum and a standard rodent diet.

**2.6.2 Assessment of acute oral toxicity.** The adoption of OECD test standards (TG) 425 enabled the assessment of acute oral toxicity in SHE (Substance Here). Initially, a single female rat was orally administered the maximum test dose (2000 mg/kg) while being closely

monitored for the initial 30 minutes [39], followed by the subsequent 4 hours. Upon survival of this rat, five additional rats were subjected to the exact dose. Simultaneously, a control group consisting of five animals received distilled water. Both treatment and control groups received equal care.

Observations were carried out meticulously at 30-minute intervals over a six-hour period, followed for 14 days. Changes in body weight and behavioral patterns were noted during these evaluations. After the 14-day period, all animals' weights were recorded, and under anesthesia, blood and serum samples were collected through heart puncture. The collected serum was then divided for biochemical tests. Subsequently, the animals were euthanized via cervical dislocation, and their heart, liver, lungs, spleen, and kidneys were extracted for histological examination after preservation in 10% formaldehyde [40].

## 2.7 Histopathological studies

Heart, liver, lungs, spleen, and kidney slices were kept in formalin solution (10%) for histological investigation. Tissues of these organs were cut into pieces with a microtome, stained along hematoxylin and cosine. After that it was observed under a light microscope to detect pathological changes [41].

## 2.8 Investigating Anti-Parkinson's activity

**2.8.1 Experimental design.** Rats were divided into six groups (n = 6) and each group containing 5 rats.

Group I: Control group (CG) (n = 5) (healthy wistar rats received vehicle)

Group II: Disease control (DC) (n = 5) was given paraquat (10 mg/kg, i.p.) [42].

Group III: Standard group (ST) (n = 5) was given L-dopa (100 mg /kg, p.o.) and carbidopa (25 mg/kg, p.o.) + paraquat (1 mg/kg, i.p.).

Group IV: Mitigated along paraquat (n = 5) (10 mg/kg, i.p.) + SHE 150 mg/kg, p.o.

Group V: Mitigated along paraquat (n = 5) (10 mg/kg, i.p.) + SHE 300 mg/kg, p.o.

Group VI: Mitigated along paraquat (n = 5) (10 mg/kg, i.p.) + SHE 600 mg/kg, p.o.

Each animal having weight (250±4) received above mentioned mitigations throughout the course of the following twenty-one days in groups. Water and food were given out in the normal way. At the beginning and end of the trial, changes in weight and behavior were noted. After administering paraquat for 60 minutes on day 21, behavioural parameters were observed in all groups excluding the control group. Rats were slaughtered at day 22 by cervical dislocation under a light anesthesia; the brains were then removed, cleaned with phosphate buffer, and kept for biochemical analyses and histological examination.

**2.8.2 Behavioral studies.** *2.8.2.1 Y-maze test*. This method was used to test rat's cognitive ability and short-term and spatial memory in behavioural neurosciences. The methods employed by Aydin et al. were largely followed in carrying out this task, the Y-maze. Short-term memory was evaluated using the Y-maze spontaneous alternation behavioural task [43]. The Y-maze used in this experiment featured three arms that were each 35 centimeters long, 25 centimeters high and 10 centimeters wide. The center section was an equilateral triangle shape. Rats were allowed unrestricted access to the maze for 8 minutes while being tethered to one arm's end. The rat was seen to have entered the arm when its hind paws were totally inside its limb. Entry into all 3 arms on subsequent judgments was deemed to indicate a spontaneous alteration in behavior. The total number of inserted arms was divided by the maximum quantity of spontaneous alternation behaviors, which was computed using the formula (actual alternations/maximum alternations) 100. Spontaneous alternation conduct is thought to be a reflection of spatial working memory, a category of short-term memory. Before putting the

next animal through the maze, the environment was cleaned by 10% ethanol solution and dried with a cloth [44].

**%age alteration** was measured by using following formula:

$$Alternation\ percentage = \frac{Actual\ alteration}{(Maximum\ spontaneous\ alteration)}X100$$

$$Spontaneous\ alyerations = Total\ number\ of\ arm\ entered - 2$$

*2.8.2.2 Hole-board test.* Non-motor symptoms including anxiety, memory loss, and sadness are also linked to PD and lower the patient's quality of life. The hole-board device is used to observe mouse behavior that resembles anxiolytic behavior. A grey, 50× 50 ×50 cm wooden box with 3 cm holes uniformly spaced around the floor to prevent escape makes up this contraption. The experimental animals were administered drugs on the twenty-first day of dosing, and after thirty minutes, each animal was put inside the apparatus and given five minutes to investigate. How many head dips took place in 5 minutes was noted. When both of the rat's eyes dropped into the hole, it was considered to be one dip. The results were presented as the mean of all head dips [44].

*2.8.2.3 Open field test.* The setup for this study included a resin-coated wooden square box that was 45 cm high, 100 cm wide, and 100 cm deep, as well as a floor that was divided into 25 squares. The following parameters were observed and noted while rats were allowed to roam freely inside the box for two minutes (1) Total line crossed (2) Time spent in central area [44].

*2.8.2.4 Wire hanging test.* This test managed to look into the animal's neuromuscular endurance and strength. It is a simple, low-cost method for evaluating rat' grip strength and neuromuscular control. The straightforward techniques for estimating muscle strength require technical expertise and more advanced tools. Despite the fact that some restrictions exist and lead to inconsistent findings, such as results that are affected by animal behavior, weight, balance, and animal handling and do not reflect the only neuromuscular impairment. The equipment utilized for this test was buildup of timber walls that were 50 cm high and three inches wide, with horizontal stainless-steel grids positioned on top. Animals were supported while being carefully handled from the tail up until their fore and hind paws grasped the grill. Furthermore, they hung on it in an erect position. For a maximum of 30 seconds, animals must remain fastened to the wire. The 30-second hanging time was increased to one minute [44].

**2.8.3 Assessment of oxidative stress biomarkers.** *2.8.3.1 Tissue homogenate preparation.* A 1:10 (w/v) ratio of phosphate buffer (pH 7.4) was utilized to homogenize the brain tissues. The homogenate was centrifuged for 10 min. at 600 rpm at 4°C, yielding a clear supernatant. A portion of this supernatant was removed and consumed for biochemical examination.

*2.8.3.2 Estimation of protein level.* After adding 4.5 ml of Reagent 1 (2% $Na_2CO_3$ in 0.1 N NaOH, 1% sodium potassium tartrate in distilled water, and 1% copper sulphate) to a test tube with 0.2 ml of brain homogenate, it was left in for 10 minutes. Following the addition of 0.5 mL of Reagent 2 (composed of 1 part H2O and 2 parts (2 N) Folin-Phenol); the solution was incubated at 37°C for 30 minutes. BSA was used as the reference to calculate the protein content after the absorbance at 660 nm was measured.

*2.8.3.3 Assay for malondialdehyde (MDA) level.* Thiobarbituric acid (TBA) solution was utilized to quantify the concentration of MDA in order to indirectly measure lipid peroxidation. The homogenate was centrifuged for five minutes at 1500 rpm to get the supernatant. One milliliter of supernatant was combined with 0.38% weight-for-weight TBA (TCA), 2.5 milliliters of 0.25 M HCL and three milliliters of trichloroacetic acid (15%). After being stirred for fifteen

minutes, cooled, and maintained for fifteen minutes, the assay mixture was centrifuged for ten minutes at 3500 rpm. At 532 nm, the obtained top layer was tested using a spectrophotometer. The entire procedure was done three times [45].

Using the following formula, the level of malondialdehyde was determined as nmoles/ mg of tissue protein:

$$\text{MDA concentration} = \frac{\text{Abs} \times 100 \times \text{volume of mixture used}}{(1.56 \times 105) \times \text{tissue weight} \times \text{Volume of aliquot used}}$$

*2.8.3.4 Assay for catalase (CAT) level.* 50 mmolar of K3PO4 buffer (1.95 mL) with a pH of 7.0 was added to 0.05 mL of brain homogenate (0.1 mL). After that, a sample mixture was combined with 1 mL of a 30 millimolar hydrogen peroxide solution, and the mixture's absorbance was measured spectrophotometrically at 240 nm. The hydrogen peroxide solution was made by mixing 0.05 mL of hydrogen peroxide with 50 mL of distilled water. A 50 mM potassium phosphate solution was prepared by mixing disodium phosphate (0.825g) and monosodium phosphate (0.265g) with 100 mL of water [45].

Following equation was used to calculate CAT level:

$$\text{CAT} = \frac{\delta \text{ O.D}}{E \times \text{Volume of sample (mL)} \times \text{protein (mg)}}$$

Were

**δOD** Change in absorbance per minute.

**V** Volume of tested sample,

**E** Extinction coefficients of $H_2O_2$

*2.8.3.5 Superoxide dismutase (SOD) activity assay.* In brief, pyrogallol (0.1 mL) and 0.1 M K3PO4 buffer (2.8 mL) were combined to yield a 3 mL combination including brain homogenate. 500 mg of pyrogallol was blended with 12 g of KOH in 8 mL of water to make a 10 mL pyrogallol solution. The assay mixture's absorbance was noted spectrophotometrically at 560 nm [45]. The level of Superoxide dismutase was assessed after the calibration curve for SOD was established using concentrations ranging from 10 L to 100 L.

*2.8.3.6 Analysis for glutathione level.* Homogenate (1 mL) was mixed with 10% TCA (trichloroacetic acid). Next is the DTNB reagent, which is created by dissolving DTNB (29.78 mg) in 25 mL of methanol. After that, a 4 mL (0.1 M) sodium phosphate buffer solution with a pH of 7.4 is added [45]. The absorbance was noted at 412 nm, and the GSH level was deliberated using an equation.

$$\text{GSH level} = Y - \frac{0.00314}{0.034} \times \frac{\text{dilution factor}}{\text{tissue homogenate}} \times \text{aliquot volume used}$$

**2.8.4 Analysis of neurotransmitter's levels.** *2.8.4.1 Preparation for aqueous phase.* The brain tissue was homogenized and centrifuged at 2000 rpm for 10 minutes in 5 mL of HCl butanol. Following centrifugation, a portion of the supernatant was combined along 0.31 mL HCl and 2.5 mL heptane, vigorously blended, then centrifuged once more at 2000 rpm for ten minutes. The mixture was centrifuged to split it into two layers, and 0.2 mL of the lower aqueous layer was used for further neurotransmitter analysis.

*2.8.4.2 Noradrenaline and dopamine levels.* 50 ml of HCl (0.4 M); 200 ml of aqueous phase; and 100 ml of EDTA solution were put in to this reaction mixture. An oxidation reaction was started by the addition of 100 ml of iodine solution. Na₂SO₃ and acetic acid were combined to the reaction mixture after 1.5 minutes to stop the oxidation. The resulting mixture will be

heated for 6 minutes at 100 oC. Dopamine's absorbance was noted at 352 nanometers, and nor-adrenaline at 452 nanometers. The blank solution for nor-adrenaline and dopamine was made by adding $Na_2SO_3$ before iodine [46].

*2.8.4.3 Measurement of serotonin levels.* Aqueous phase and O-phthaldialdehyde were combined and heated at 100 oC for 10 minutes to estimate serotonin levels. As a control, HCl was utilized. At 440 nm, the sample solution's and the blank solution's absorbance was noted.

*2.8.4.4 Evaluation of Acetyl Cholinesterase (AChE) activity.* 2.6 mL of phosphate buffer (0.1 M; pH 8); supernatant was combined to 0.4 ml of the buffer. Then it was mixed with DTNB (0.1 ml) and acetylthiocholine iodide (20 ml). As a result of the chemical interaction between DTNB and acetylthiocholine iodide, a 412 nanometer yellow color was produced. Until the 10 minutes were up, the absorbance was measured every two minutes. The measure of enzyme activity was μM per minute per mg of tissue.

**2.8.5 Histopathological studies.** After being removed by sacrifice, the brain was kept in a 10% formalin solution while being fixed in paraffin wax blocks. For histopathological research, silver staining was carried out and examined with a 10X light microscope equipped with a camera using eosin and Hematoxylin, or H&E, dyes.

**2.8.6 Estimation of inflammatory biomarkers by ELISA.** The elabscience ELISA kit for measuring IL-6(Catalog No: E-ELR0015) and TNF-α (Catalog No: E-ELR0019) was utilized, and the product's instructions were followed.

**2.8.7 qRT-PCR.** Rat's brain hemispheres were used for RNA extraction, initially homogenized with a Polytron device, followed by treatment with Trizol from Life Technologies, Carlsbad, CA, USA. Subsequently, the RNA samples underwent cDNA synthesis in a 20 μL volume using the QuantiTect reverse transcription kit by Qiagen. qRT-PCR was carried out with the following thermal cycling conditions: an initial denaturation step at 95˚C for 5 minutes, followed by 40 cycles consisting of denaturation at 95˚C for 15 seconds, annealing at 60˚C for 20 seconds, and extension at 72˚C for 20 seconds [41].

qRT-PCR was employed to assess mRNA expression levels of various markers, using GADPH as an internal reference (Table 1). The Mesa Blue qRT-PCR Master Mix Plus for the SYBR assay from Eurogentec was utilized on the Master cycler Realplex2 by Eppendorf. Relative quantitation was determined using the comparative threshold cycle ($C_T$) method with realplex software. The mean $C_T$ values from triplicate measurements were used to calculate $\Delta C_T$, representing the difference in $C_T$ between the target genes and the internal reference (GADPH). $\Delta\Delta C_T$ was then computed by finding the difference between the $\Delta C_T$ of the control

**Table 1. List of primers used in qRT-PCR analysis of paraquat induced Parkinson's disease rodent model.**

| Forward/Reverse | Biomarkers | Sequence | Accession No. |
|---|---|---|---|
| **Forward** | IL-1β | GACTTCACCATGGAACCCGT | NM_031512.2 |
| **Reverse** | IL-1β | GGAGACTGCCCATTCTCGAC | |
| **Forward** | AchE | TAGCACCCCACTCCATTCTCA | NM_172009.1 |
| **Reverse** | AchE | TCCCCTCAACATCAGGCTCA | |
| **Forward** | TNF-α | GGAGGGAGAACAGCAACTCC | NM_012675.3 |
| **Reverse** | TNF-α | TCTGCCAGTTCCACATCTCG | |
| **Forward** | α-Synuclein | TCGAAGCCTGTGCATCTCG | XM_017592500.1 |
| **Reverse** | α-Synuclein | CTCCCTCCTTGGCCTTTGAA | |
| **Forward** | GAPDH | GGAGTCCCCATCCCAACTCA | XM_0175922435.1 |
| **Reverse** | GAPDH | GCCCATAACCCCCACAACAC | |

Following qRT-PCR, the PCR products from tissue samples were subjected to electrophoresis using the E-Gel Precast Agarose Electrophoresis System.

experiment and that of each sample. The fold increase in mRNA was determined using the formula $2^{-\Delta\Delta C_T}$ [41].

**2.8.8 Statistical analysis.** GraphPad Prism version 5 was used to statistically analyze the data, and mean was utilized to express each value along with standard error of measurement (SEM). One-way or two-way ANOVA, followed by the Tukey's post-test for multiple comparisons.

# 3. Results

## 3.1 Estimation of secondary metabolites from plant extract

Among secondary metabolites percentage of total polyphenolic content is higher i.e. 8.50 as compared to total flavonoid content i.e. 3.02 in ethanolic extract of *Syzygium heyneanum*. For the quantification of total flavonoid content, regression equation y = 0.0036x - 0.078 with $R^2$ = 0.9887 was acquired from the standard curve of quercetin, while regression equation y = 0.0064x + 0.04 with $R^2$ = 0.9904 was come by the standard curve of gallic acid.

## 3.2 In vitro antioxidant activity

DPPH scavenging assay shows maximum percentage inhibition 97.85 at lower SHE concentration while minimum inhibition is 88.26 at higher SHE concentration with $IC_{50}$ value 42.13 for all SHE concentrations and percentage inhibitions. Results are exhibited in Table 2.

## 3.3 HPLC analysis

HPLC analysis finds out 6 compounds with their names and retention time (Table 3). According to HPLC chromatogram (Fig 1) Caffeic acid, Vinallic acid, Sinapic acid and Benzoic acid have moderate peaks while P-coumaric acid and Benzoic acid have sharp peaks with retention time 3.508, 14.891 respectively.

## 3.4 Acute oral toxicity study

**3.4.1 Histopathological analysis.** The objective is to detect any potential abnormalities or toxicity in the treatment group versus the control. The histopathological analysis of SHE does not show any toxic effects. Results are mentioned in Fig 2.

**Table 2. Determination of antioxidant activity of SHE extract through DPPH assay.**

| Sr. No. | Ascorbic acid concentration | percentage Inhibition | SHE concentration | percentage Inhibition |
|---|---|---|---|---|
| 1 | 0 | 0 | 0 | 0 |
| 2 | 18 | 20 | 7 | 28 |
| 3 | 28 | 28 | 14 | 32 |
| 4 | 35 | 30 | 24 | 43 |
| 5 | 48 | 48 | 33 | 49 |
| 6 | 50 | 50 | 42 | 50 |
| 7 | 60 | 50 | 50 | 51 |
| 8 | 68 | 55 | 57 | 52 |
| 9 | 74 | 62 | 67 | 62 |
| 10 | 80 | 65 | 70 | 64 |
| 11 | 88 | 70 | 76 | 81 |
| 12 | 95 | 83 | 88 | 93 |
| $IC_{50}$ 16.24 | | | **$IC_{50}$ 42.13** | |

Column 1 shows ascorbic acid concentration gradient which served as standard.

**Table 3. Phytochemicals detected in Hplc analysis of *S. heyneanum* extract with retention time.**

| Sr. no | Retention time (min) | Compound detected |
|--------|----------------------|-------------------|
| 1 | 3.508 | P-coumaric acid |
| 2 | 6.941 | Chlorogenic acid |
| 3 | 11.405 | Caffeic acid |
| 4 | 13.420 | Vinallic acid |
| 5 | 13.861 | Sinapic acid |
| 6 | 14.891 | Benzoic acid |

## 3.5 Investigation of antiparkinson's activity

**3.5.1 Behavioral studies.** *3.5.1.1 Y-maze test.* The exploration and cognitive deficiency are properly tested using the Y-maze test. The study's findings demonstrate that the illness control group's percent alteration and number of arm entries were considerably lower ($P < 0.05$) than those of the normal control group. The percentage change and the overall number of arm entries both demonstrate significant ($P < 0.05$) outcomes for the SHE treated groups.

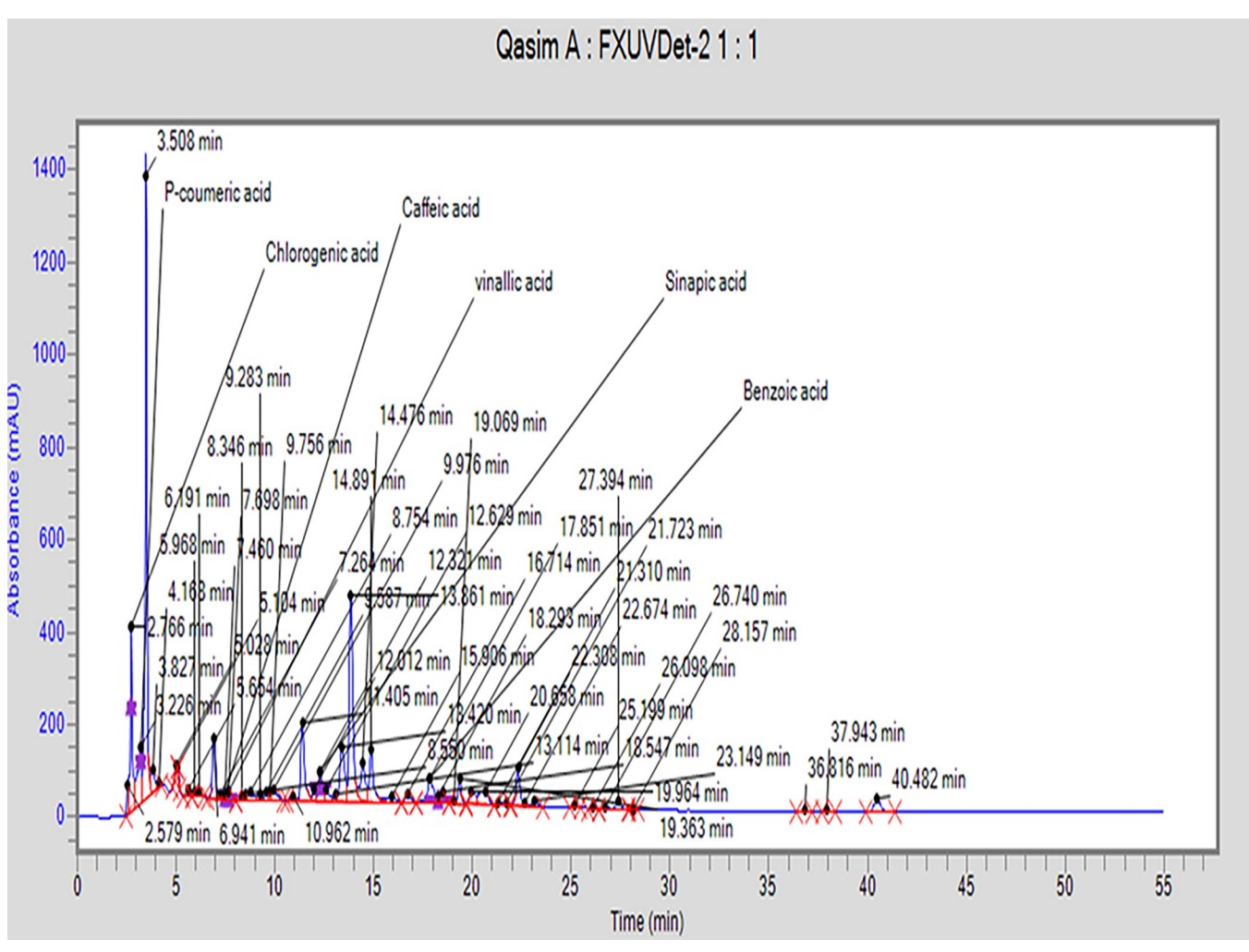

**Fig 1. Representation of Hplc chromatogram of *S. heyneanum* extract.**

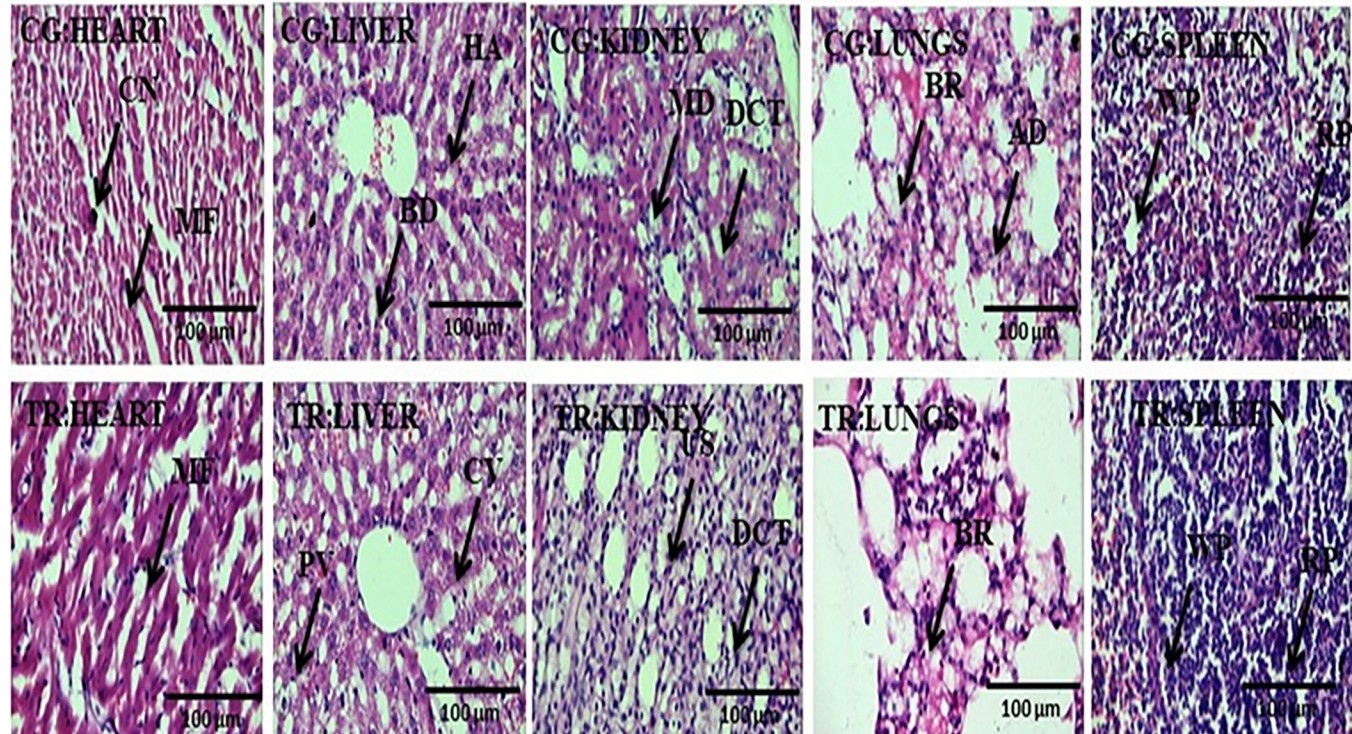

**Fig 2. Representative photomicrographs (60X) included the liver, lungs, heart, kidney, and spleen in both control (CG) and treatment (TR) groups.** Key structures like central vein (CV), portal vein (PV), bile duct (BD), hepatic artery (HA), bronchiole (BR), alveolar duct (AD), myocardial fibrils (MF), central nuclei (CN), urinary space (US), distal convoluted tubules (DCT), proximal convoluted tubules (PCT), white pulp (WP), and red pulp (RP) were scrutinized. Encouragingly, the results indicated no significant toxic effects in these organs in the treatment group compared to the control group.

According to the provided dose level, the results demonstrated the greatest memory recovery at a treatment level of 600 mg/kg (Table 4).

*3.5.1.2 Hole-board test.* In contrast to the normal control group, the paraquat-treated group (DC) displayed significantly fewer instances of head dipping, edge sniffing, walking, and

**Table 4. Effect of SHE on neurobehavioral tests.**

| Groups | Y-maze test | | Hole-board test | Open field test | | Wire hanging test | |
|---|---|---|---|---|---|---|---|
| | Total no. of arm entries | % age alterations | Number of head dipping's | Total line crossed | Time spent in central areas (sec) | Body weight (g) | Hang time (sec) |
| CG | 12±0.577 | 39±0.577 | 21.6±0.66 | 20.3±0.33 | 53±1.52 | 159±2.4 | 58±0.577 |
| DC | 2.3±0.3[###] | 2±1[###] | 3.3±0.33[###] | 8±0.57[###] | 3.3±0.33[###] | 154±0.2 | 17.3±0.333[###] |
| ST | 13±0.577[***] | 38.6±0.882[***] | 19±0.57[**] | 25.3±0.88[***] | 52.6±0.88[***] | 173.5±2.3 | 44±1.000[***] |
| SHE 150mg/kg | 8.3±0.667[***] | 21±0.577[***] | 8±0.57[***] | 15±0.57[***] | 19±0.57[***] | 164±3.4 | 18.3±0.882[ns] |
| SHE 300mg/kg | 7.3±0.3[***] | 33.6±0.882[***] | 19.6±0.88[***] | 17.3±0.88[***] | 30±1.15[***] | 165±8.1 | 23.3±0.333[***] |
| SHE 600mg/kg | 17±0.577[***] | 39.6±0.3[***] | 23±0.57[***] | 23.6±0.88[***] | 47±1.15[***] | 158±3.4 | 59±0.577[***] |
| **F-value** | 94.08 | 403.9 | 168.5 | 77.26 | 397.2 | 45.2 | 852.9 |

The data are presented as means ± standard error of the mean (SEM) with a sample size of n = 6. Statistical significance is denoted by asterisks

([***] $p < 0.05$) when compared to the DC group, and pound signs

([###]) when compared to the CG group. Non-significant differences are indicated as 'ns'.

immobility. In treatment groups, instances of head dipping, edge sniffing, walking, and immobility are ($p < 0.05$) significantly improved (Table 4).

*3.5.1.3 Open field test.* An open-field test was used to assess the protective effects of *Syzygium heyneanum* leaf extract (SHE) on mobility, anxiety, and exploration. Parameters captured by open field equipment, such as the number of crossed lines, time spent in central area, and how often they are groomed. The overall number of squares crossing was significantly ($p < 0.05$) lower in the DC group in contrast to the CG. The standard group's square count increased significantly ($p < 0.05$). Groups exposed to the SHE treatment displayed an increase in total line crossings and time spent in central area (Table 4).

*3.5.1.4 Wire hanging test.* Wire hanging test was used to explore the latency or fall-of time which reduced significantly in the DC group in comparison to the CG and fall-of time was improved significantly in dose dependent manner and ST group when compared to the DC group. In this study SHE 300 and 600mg/kg strengthen neuromuscular coordination by improving the hang time than SHE 150mg/kg (Table 4).

## 3.6 Biochemical parameters evaluation

Paraquat treated group significantly decreased level of SOD, CAT and GSH as compared to normal CG and significantly($p < 0.05$) increased level of MDA. Treatment with SHE low, medium and high dose significantly increased level of SOD, CAT and GSH and significantly ($p < 0.05$) decreased the level of MDA (Table 5).

## 3.7 Estimation of neurotransmitters in brain homogenate

The primary neurotransmitters linked to cognitive abilities and the basal ganglia that control motor coordination include dopamine, noradrenaline, and serotonin. Neurotransmitters level decrease remarkably ($p < 0.05$) in DC as compared to CG and ST, while high dose SHE treatment has significantly ($p < 0.05$) higher level of neurotransmitters in contrast to DC. The acetylcholinesterase level of disease control (DC) is significantly ($p < 0.05$) higher due to impairment in choline acetyltransferase in contrast to control group (CG) and standard treatment (ST), while AchE level of SHE 600mg/kg (high dose) is remarkably ($p < 0.05$) lower in contrast to DC (Table 6).

## 3.8 Estimation of acetylcholinesterase

The acetylcholinesterase level of disease control (DC) is significantly ($p < 0.05$) higher due to impairment in choline acetyltransferase in contrast to control group (CG) and standard

**Table 5. Effect of SHE on first line antioxidant enzymes in brain homogenate.**

| Groups | MDA (TBA mg/ml) | CAT (IU/µl) | SOD (IU/µl) | GSH (µg/mg protein) |
|---|---|---|---|---|
| CG | 2.600±0.040 | 0.973±0.023 | 0.397±0.009 | 0.617±0.018 |
| DC | 4.437±0.143[###] | 0.740±0.015[###] | 0.073±0.003[###] | 0.223±0.009[###] |
| ST | 3.143±0.068[***] | 0.807±0.015[ns] | 0.330±0.006[***] | 0.417±0.012[***] |
| SHE 150mg/kg | 3.027±0.027[***] | 0.910±0.010[***] | 0.123±0.009[**] | 0.343±0.018[***] |
| SHE 300mg/kg | 2.753±0.073[***] | 0.950±0.006[***] | 0.217±0.012[***] | 0.430±0.012[***] |
| SHE 600mg/kg | 2.543±0.066[***] | 1.007±0.009[***] | 0.390±0.006[***] | 0.630±0.017[***] |
| **F-value** | 80.07 | 49.22 | 308.4 | 116.6 |

The data are presented as means ± standard error of the mean (SEM) with a sample size of n = 6. Statistical significance is denoted by asterisks

(*** $p < 0.05$) when compared to the DC group, and pound signs

(###) when compared to the CG group. Non-significant differences are indicated as 'ns'.

**Table 6. Effect of SHE on neurotransmitters level in brain tissue homogenate.**

| Groups | Dopamine (µg/ µg of bt) | Noradrenaline (µg/ µg of bt) | Serotonin (µg/ µg of bt) |
|---|---|---|---|
| CG | 0.334±0.017 | 0.056±0.004 | 0.637±0.026 |
| DC | 0.049±0.029### | 0.047±0.005 | 0.323±0.029### |
| ST | 0.541±0.031*** | 0.071±0.004 | 0.553±0.043** |
| SHE 150mg/kg | 0.327±0.031*** | 0.033±0.003 | 0.338±0.028ns |
| SHE 300mg/kg | 0.374±0.032*** | 0.035±0.001 | 0.368±0.023ns |
| SHE 600mg/kg | 0.538±0.017*** | 0.053±0.003 | 0.561±0.020*** |
| **F-value** | 49.01 | 14.53 | 22.16 |

The data are presented as means ± standard error of the mean (SEM) with a sample size of n = 6. Statistical significance is denoted by asterisks

(*** $p < 0.05$) when compared to the DC group, and pound signs

(###) when compared to the CG group. Non-significant differences are indicated as 'ns'.bt (brain tissue)

treatment (ST), while AchE level of SHE 600mg/kg (high dose) is remarkably ($p < 0.05$) lower in contrast to DC (Table 7).

## 3.9 Estimation of protein level

Disease control group (DC) has significant ($p < 0.05$) lower level of proteins as compared to normal control group (healthy wistar rats). Standard treatment and SHE treatment have significantly ($p < 0.05$) higher proteins level as compared to DC (Table 7).

## 3.10 Histopathological analysis of brain

Histopathological analysis revealed that neurofibrillary tangles and neuronal loss mitigated by SHE treated group dose dependently, compared to disease control group (Fig 3).

## 3.11 Estimation of inflammatory biomarkers by ELISA

Disease control group (DC) has significant ($p < 0.05$) higher level of TNF-α and IL-6 as compared to normal control group (healthy wistar rats). Standard treatment and SHE treatment have significantly ($p < 0.05$) lower TNF-α and IL-6 level as compared to DC (Table 7).

**Table 7. SHE effects on acetylcholinesterase and protein levels as well as inflammatory biomarkers by ELISA.**

| Groups | Acetylcholinesterase (µmoles/mg of proteins) | Protein level (µg/mg) | Inflammatory biomarkers | |
|---|---|---|---|---|
| | | | TNF-α level | IL-6 level |
| CG | 4.756±0.634 | 556.843±18.20 | 1058.3±22.048 | 420.9±13.051 |
| DC | 13.364±0.730### | 404.041±8.84### | 3873.3±81.921### | 559.1±12.921### |
| ST | 4.067±0.601*** | 515.551±4.92*** | 1823.3±39.299*** | 146.7±0.088*** |
| SHE 150mg/kg | 11.193±0.742ns | 477.540±6.67*** | 3034.6±21.835*** | 479.9±9.292** |
| SHE 300mg/kg | 8.377±0.652** | 509.978±4.37*** | 2048±21.656*** | 344±18.699*** |
| SHE 600mg/kg | 6.376±0.557*** | 520.186±2.50*** | 1043.000±23.072*** | 170.863±7.470*** |
| **F-value** | 31.48 | 32.50 | 738.2 | 201.3 |

The data are presented as means ± standard error of the mean (SEM) with a sample size of n = 6. Statistical significance is denoted by asterisks

(*** $p < 0.05$) when compared to the DC group, and pound signs

(###) when compared to the CG group. Non-significant differences are indicated as 'ns'.

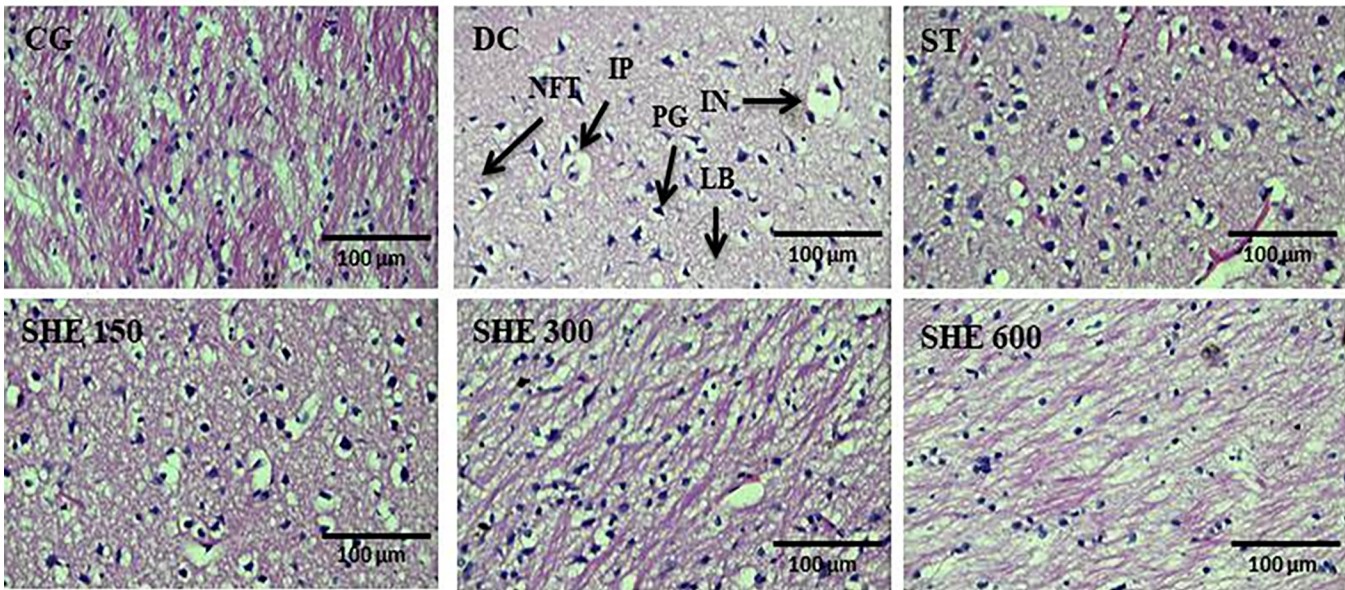

**Fig 3. Representative photomicrographs (60X), Hematoxylin and eosin staining of transverse brain sections in which disease group (DC) specify NFT: Neurofibrillary tangles, IP: intracellular plaques, IN: Inflammation, PG: Pigmentation, LB: Lewy bodies, in standard treated group (ST) neurofibrillary tangles and neuronal loss was minimum, in low dose SHE treated group (150mg/kg) neuronal loss was ameliorated.** Neuronal loss was improved in medium dose SHE treated group (300mg/kg), significantly improved architecture of high dose SHE treated group (600mg/kg) transverse brain sections.

### 3.12 qRT-PCR

Fig 4 reveald that the mRNA expression of neurodegenerative biomarkers was significantly upregulated in disease control (DC) group acetylcholinesterase, tumor necrosis factor α (TNF-α), interlukin 1b and alpha synuclein compared to control group. However, treatment with SHE dose dependently especially at higher dose 600 mg/kg significantly($p < 0.05$) downregulated the mRNA expression compared to disease control group.

## 4. Discussion

Parkinson's disease (PD) is a clinical malady that can be assessed and has a number of different etiologies and clinical symptoms. Except for viral origins, it is a neurodegenerative ailment with a rising frequency in all countries. Due to the lack of a particular treatment, rising morbidity and high costs of pharmaceuticals for neurodegenerative illnesses, research into novel herbal neuroprotective chemicals is crucial [47]. As a result of, the presence of secondary metabolites like alkaloids, flavonoids and polyphenols, phytochemical investigations of *S. heyneanum* indicated its medicinal significance. Nutritional supplements and herbal extracts high in flavonoids have been showing improve in memory, cognitive function, and learning while also slowing neurodegeneration. Flavonoid-rich plant extracts are advantageous for treating neurodegenerative diseases as a result of their alteration of intracellular signalling pathways, which enhance cell survival and age-related neural activity [48]. *S. heyneanum* is rich in flavonoids and polyphenols, which can be used as a base to create new neuroprotective drugs because they are strong antioxidants with anti-ischemic actions.

Increasing neuronal regeneration and lowering oxidative stress are the main ways flavonoids exert their neuroprotective benefits [49]. Therefore, the primary ingredients for the creation of a neuroprotective agent are the flavonoids found in *S. heyneanum* extract. Our results from DPPH and HPLC study demonstrated *S. heyneanum* potential for neuroprotection and

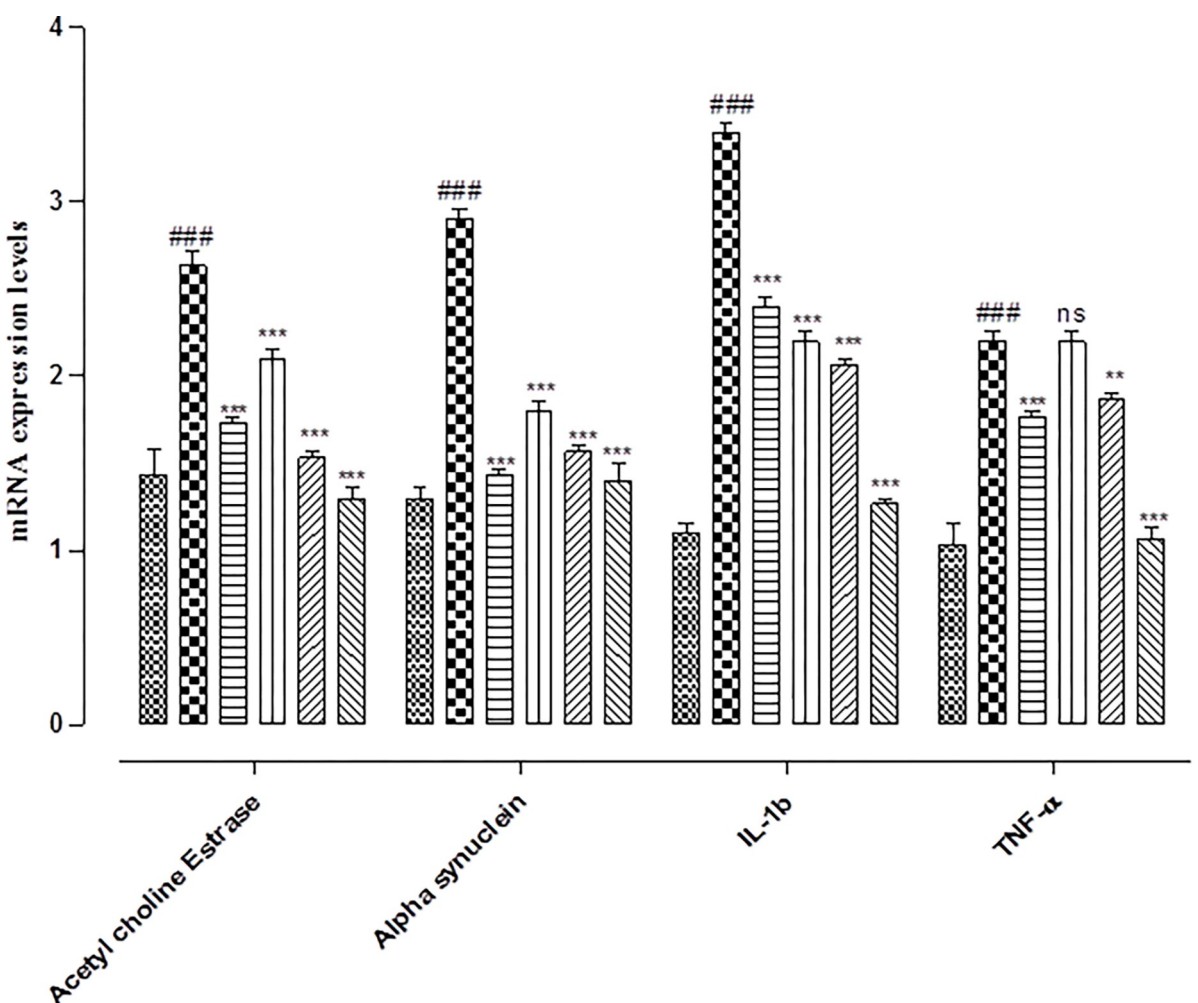

**Fig 4. Effect of SHE on mRNA expression of principal biomarkers inculpated in pathology of PD, data is presented as mean ± SEM (*n* = 6).** Statistical significance is presented by *** (p < 0.05) in contrast to the DC group, ### compared to CG, while non-significant differences are indicated as ns.

antioxidant activity, and HPLC examination revealed the presence of six distinct compounds. They mostly consist of phenolic and flavonoid chemicals. The strongest peaks are found in phenolic acids for instance p-coumaric, Chlorogenic, as well as Caffeic acid, which are the best antioxidant chemicals.

Treatment dosages demonstrated dose-dependent behavior, SHE at greatest dose of 600 mg/kg produced best results. The results of the study show that the dopamine level in the disease control group (DC) was significantly lower compared to the normal control group (CG) and the standard treatment (ST) group. However, treatment with SHE at different doses significantly increased the level of dopamine in a dose-dependent manner when compared to the DC group. This suggests that SHE has a potential positive impact on the dopamine neurotransmitter system, which is associated with cognitive function and motor coordination, particularly in the context of conditions like Parkinson's disease. While MDA levels fell in contrast to the DC group, SOD, CAT, and GSH proportions become greater [50].

Improvement in both motor and non-motor symptoms was determined using behavioural tests like the Y maze, wire hanging, open field, and hole board tests. The assessment of neurotransmitters yielded favorable findings as well. The proportions of dopamine, serotonin, and nor-adrenaline and acetyl cholinesterase significantly increased, when the levels were comparable to DC. Oxidative stress, which is due to the creation of ROS, lowers the protein level in the disease control model. According to the study, protein levels in patients receiving a high dose of SHE (600 mg/kg) compared to disease control group, significantly increased ($p < 0.05$).

Additionally, histological inspections was carried out to examine the acute toxicity of the SHE at 1000 mg/kg on essential organs such the lungs, liver, heart, spleen, and kidney as the disease group displayed normal behavior for both the control and treatment groups. In order to designate safer dose limits, detect clinical signs and symptoms, and ascertain the therapeutic index of herbal medications, acute toxicity studies are necessary. When used excessively on vital organs, herbal remedies that are therapeutically effective can have negative side effects.

An increase in the appearance of proinflammatory cytokines like TNF-α and IL-6 is closely correlated with the development of Parkinson's disease, according to prior research. Increased levels of TNF-α have the potential to increase the levels of certain cytokines that may contribute to neurodegeneration. Proinflammatory cytokines (TNF-α and IL-6) are responsible for the dopaminergic cell loss in the PD. They are potent microglial function mediators that contribute to the death of dopaminergic neuronal cells. According to recent studies, anti-inflammatory substances found in plants can be used to treat neurodegenerative illnesses. The IL-6 increased the proportions of the MAPKA pathway and raised the amount of NF-KB, which led to neuro-inflammation [33]. Due to brain injury in the basal ganglia region, TNF-α and IL-6 levels are higher in the DC group. On the level of inflammatory biomarkers, the SHE had decreasing effects that were dose dependant. The SHE 600 mg/kg dosing group showed the most notable ($p < 0.05$) effect.

Acetylcholinesterase (AChE) is primarily responsible for stopping cholinergic neurotransmission at synapses by hydrolyzing acetylcholine [51]. Dopaminergic neuronal cell death is connected to elevated AChE mRNA expression. In a paraquat-induced animal model, the neurotoxic effect increased AChE levels and its mRNA expression. SHE has the potential to treat apoptotic dopaminergic neuronal cell death because it inhibits AChE expression in contrast to DC.

By activating microglial cells in the striatum and CSF, PD etiopathogenesis is strongly correlated with increased expression of proinflammatory cytokines including TNF-α and IL-1β. Dolrahman et al. find out that P-coumaric acid have ability to reduce the levels of malondialdehyde and TNF-α in rotenone induced PD models and it also provide protection from neuronal loss in the SNc and striatum [52]. Similarly HPLC studies of *S.heyneanum* revealed that P-coumaric acid is present in high ppm. Through the regulation of neuroinflammation and oxidative stress, SHE reduced the higher levels of the proinflammatory cytokines TNF-α and IL-1β in treatment groups compared to the DC group.

The key neuropathological hallmark of PD is the accumulation of α-synuclein protein in Lewy bodies and Lewy neuritis, as well as the death of dopaminergic neurons in the substantia nigra. Presynaptic protein α-synuclein, which has 140 amino acids, coordinates the release of neurotransmitters from synaptic vesicles in the central nervous system. Increased amounts of α-synuclein protein brought on by SNCA gene mutations play a significant role in the pathogenetic dysfunctions of PD [53]. Chlorogenic acid was found by Xin et al. to have neuroprotective and antioxidant potential that treated the 1-methyl-4-phenyl-1-1,2,3,6-tetrahydropyridine (MPTP) zebra fish model of PD by modulating the expression of PD (α-synuclein), demonstrating its ability to promote the autophagy that was disrupted in the PD pathology. As

chlorogenic acid is a main phytochemical constituent of SHE, so it have great impact on $\alpha$ – synuclein expressions [54].

It has been discovered that the bulk of the phytoconstituents stimulate neuroprotective, regulated cell-stress response pathways. According to our findings, SHE acts as a modulator against the behavioural and physiological alterations caused by Paraquat-induced neurotoxicity. Furthermore, it has the ability to slow the progression of PD and associated symptoms.

## 5. Conclusion

According to the results of the current comprehensive study, PD is prevented by *Syzygium heyneanum* antioxidant and neuroprotective effects. Due to the inclusion of polyphenols and flavonoids, SHE was able to considerably improve the biochemical, histological, and neuromuscular parameters. In addition, ELISA investigations, neurotransmitter estimate, and qRT-PCR analysis validated the use of extract in PD and provided fresh information for the creation of innovative treatments for PD from natural sources that might be beneficial to the body.

## Author Contributions

**Conceptualization:** Malik Saadullah, Hafsa Tariq, Zunera Chauhdary, Uzma Saleem, Shazia Anwer Bukhari, Muhammad Asif, Aisha Sethi.

**Data curation:** Malik Saadullah, Hafsa Tariq, Zunera Chauhdary, Amna Sehar.

**Formal analysis:** Malik Saadullah, Hafsa Tariq, Uzma Saleem, Shazia Anwer Bukhari, Amna Sehar, Aisha Sethi.

**Investigation:** Zunera Chauhdary, Shazia Anwer Bukhari, Amna Sehar, Muhammad Asif.

**Methodology:** Zunera Chauhdary, Uzma Saleem, Shazia Anwer Bukhari, Muhammad Asif.

**Project administration:** Muhammad Asif.

**Resources:** Uzma Saleem.

**Supervision:** Malik Saadullah, Shazia Anwer Bukhari, Aisha Sethi.

**Validation:** Shazia Anwer Bukhari, Aisha Sethi.

**Visualization:** Hafsa Tariq, Shazia Anwer Bukhari, Aisha Sethi.

**Writing – original draft:** Malik Saadullah, Hafsa Tariq, Zunera Chauhdary.

**Writing – review & editing:** Malik Saadullah, Hafsa Tariq, Zunera Chauhdary, Amna Sehar.

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
