## [Decision Letter · Decision Letter 0]

24 Nov 2023

PONE-D-23-32989Biochemical properties and biological potential of Syzygium heyneanum with Antiparkinson’s activity in paraquat induced rodent modelPLOS ONE

Dear Dr. Chauhdary,

Thank you for submitting your manuscript to PLOS ONE. After careful consideration, we feel that it has merit but does not fully meet PLOS ONE’s publication criteria as it currently stands. Therefore, we invite you to submit a revised version of the manuscript that addresses the points raised during the review process. Please submit your revised manuscript by Jan 08 2024 11:59PM. If you will need more time than this to complete your revisions, please reply to this message or contact the journal office at plosone@plos.org. Please include the following items when submitting your revised manuscript:A rebuttal letter that responds to each point raised by the academic editor and reviewer(s). You should upload this letter as a separate file labeled 'Response to Reviewers'.A marked-up copy of your manuscript that highlights changes made to the original version. You should upload this as a separate file labeled 'Revised Manuscript with Track Changes'.An unmarked version of your revised paper without tracked changes. You should upload this as a separate file labeled 'Manuscript'.

We look forward to receiving your revised manuscript.

Kind regards,

Yasmina Abd‐Elhakim

Academic Editor

PLOS ONE

Journal Requirements:

**Reviewers' comments:**

Reviewer's Responses to Questions

**Comments to the Author**

1. Is the manuscript technically sound, and do the data support the conclusions?

Reviewer #1: Yes

Reviewer #2: Partly

2. Has the statistical analysis been performed appropriately and rigorously? 

Reviewer #1: Yes

Reviewer #2: N/A

3. Have the authors made all data underlying the findings in their manuscript fully available?

Reviewer #1: Yes

Reviewer #2: No

4. Is the manuscript presented in an intelligible fashion and written in standard English?

Reviewer #1: Yes

Reviewer #2: No

**5. Review Comments to the Author**

**Reviewer #1:** The manuscript entitled “ Biochemical properties and biological potential of Syzygium heyneanum with Antiparkinson’s activity in paraquat induced rodent model” aimed to evaluate the main components of Syzygium heyneanum and its potential protective role against Parkinson’s disease. The manuscript contains a good employment for laboratory techniques to evaluate the components of Syzygium heyneanum. However, the experimental design doesn’t contain very important information about the animals used in the experiment “ weight , age , adaptation period , ethical code statement , the source of the animals , food and water , humidity and other environmental conditions, How??

Some minor points are required:

- Replace “ RT-PCR” by “ qRT-PCR”

- Several sites in the introduction need citation support for eg “ Consequently a potent complementary herbal medicinal agent is a vital clinical requirement in the modern world'. Cite the relevant reference in this regard.

- The aim of the study needs revision , although it is focused but it needs to illustrate more on the consequences of S. heyneanum against PD.

- The method of plant extraction in the methods section needs reference.

- The subsections are not well defined.

- The formula for estimating the total flavonoids, and total phenolic compounds should be explained and references should be added.

- What do you mean by “followed for 14 days”? , do you mean inspected ?

- The exact name of the method used for detection of noradrenaline and serotonin should be frankly written and the specified reference you followed should be added.

- “ Estimation of protein level” should be after the method of homogenate preparation.

- The method of QRT-PCR : is very pably presented and contains very little information about the PCR conditions, RNA extraction and the used kits or the method of fold change classification .

- How could you confirmed the primers used? What are the basis of selecting Gapdh as a normalizing gene?

- Remove table (2) and add these information only in the text , it contains just one raw.

- Table 3: Explain the content in the footnote of the table, what you men by the first column ? are these concentration gradient ?

- What does this mean “ %age Inhibition”?

- Table 4 : is not cited in the text and all its content should be illustrated in the results text.

- The format of the table 5, 6 , 7 and 8 is so confusing and not well presented , please , reconfigure these tables and collect them in a single table contains these data. And so reverse all other presented tables revise and uniform when possible.

- Specify the sections for histopathological examination in brain tissue , add the score of different detected lesions.

**Reviewer #2:** Authors revised the manuscript partially, still few improvement required.

1. Suggested to perform one-way ANOVA, authors stated that they have performed. But the information such as F- value, which post-hoc analysis carried out is missing.

2. The Table-2, reporting the change in the protein level between the experimental group. This analysis is very basic, measuring the total protein content is not considered as research data. This table should be removed.

3. Figure 2 & 3, the immunohistochemistry data is not clear, provided image is only 10X. At least, to see some details the magnification should be 40X or 60X.

6. PLOS authors have the option to publish the peer review history of their article (what does this mean?). If published, this will include your full peer review and any attached files.

Reviewer #1: No

Reviewer #2: No

---

## [Author Response · Author response to Decision Letter 0]

14 Dec 2023

Response to Reviewers

Reviewers'comments:

Reviewer #1: The manuscript entitled “ Biochemical properties and biological potential of Syzygium heyneanum with Antiparkinson’s activity in paraquat induced rodent model” aimed to evaluate the main components of Syzygium heyneanum and its potential protective role against Parkinson’s disease. The manuscript contains a good employment for laboratory techniques to evaluate the components of Syzygium heyneanum. However, the experimental design doesn’t contain very important information about the animals used in the experiment “ weight , age , adaptation period , ethical code statement , the source of the animals , food and water , humidity and other environmental conditions, How??

Answer: The information regarding animal age, sex, environmental conditions, acclimation, food and water accessibility and source are incorporated in the revised manuscript. 

Some minor points are required:

- Replace “ RT-PCR” by “ qRT-PCR”

Answer: This comment has been addressed by replacing all RT-PCR by qRT-PCR.

- Several sites in the introduction need citation support for eg “Consequently a potent complementary herbal medicinal agent is a vital clinical requirement in the modern world'. Cite the relevant reference in this regard.

Answer: All the references are cited properly in revised manuscript. 

- The aim of the study needs revision , although it is focused but it needs to illustrate more on the consequences of S. heyneanum against PD.

Answer: This section is revised and incorporated in revised manuscript. 

- The method of plant extraction in the methods section needs reference.

Answer: References are added in revised manuscript

- The subsections are not well defined.

Answer: The subsections has been properly described in revised manuscript.

- The formula for estimating the total flavonoids, and total phenolic compounds should be explained and references should be added.

Answer: Formulas are incorporated in revised manuscript. 

- What do you mean by “followed for 14 days”? , do you mean inspected ?

Answer: Acute toxicity study was conducted according to OECD 425 guidelines, treatment was continued for 14 days. During and after that animals were observed for any abnormal behavior and vital organs were removed to determine any lethal effect at completion of 14 days. 

- The exact name of the method used for detection of noradrenaline and serotonin should be frankly written and the specified reference you followed should be added.

Answer: The level of neurotransmitters was determined by calorimetric method from brain homogenate and the following reference is incorporated in the manuscript. 

Waseem, Wajeeha, Fareeha Anwar, Uzma Saleem, Bashir Ahmad, Rehman Zafar, Asifa Anwar, Muhammad Saeed Jan, Umer Rashid, Abdul Sadiq, and Tariq Ismail. "Prospective Evaluation of an Amide-Based Zinc Scaffold as an Anti-Alzheimer Agent: In Vitro, In Vivo, and Computational Studies." ACS omega 7, no. 30 (2022): 26723-26737.

- “ Estimation of protein level” should be after the method of homogenate preparation.

Answer: “Estimation of protein level” incorporated after homogenate preparation in revised manuscript.

- The method of QRT-PCR : is very pably presented and contains very little information about the PCR conditions, RNA extraction and the used kits or the method of fold change classification .

Answer: the detail of all these sections with kits detail are incorporated in revised manuscript. 

- How could you confirmed the primers used? What are the basis of selecting Gapdh as a normalizing gene?

Answer: The used primers were confirmed by BLAST Analysis and by primer design tools which predict secondary structures, melting temperatures, and primer-dimer formation. Primers were also experimentally validated by the PCR product on an agarose gel to ensure the expected size and absence of nonspecific bands and by melting curve analysis. Selecting GAPDH (Glyceraldehyde 3-phosphate dehydrogenase) as a normalizing gene, also known as a housekeeping gene, is selected due to its ubiquitous expression involved in a fundamental metabolic pathways of all tissues and cells. Under normal physiological conditions, GAPDH expression remains relatively stable, making it a suitable reference gene for comparing gene expression levels among different samples. Therefore, according to previous references we prefer to use GADPH as normalizing gene. 

- Remove table (2) and add these information only in the text , it contains just one raw.

Answer: Table 2 adjusted as suggested in revised manuscript

- Table 3: Explain the content in the footnote of the table, what you men by the first column ? are these concentration gradient ?

Answer: Table 3 adjusted as suggested in revised format. 

- What does this mean “ %age Inhibition”?

Answer: This means percentage of Inhibition, revised in main manuscript format. 

- Table 4: is not cited in the text and all its content should be illustrated in the results text.

Answer: Table 4 is cited in the text in results sections. 

- The format of the table 5, 6 , 7 and 8 is so confusing and not well presented , please , reconfigure these tables and collect them in a single table contains these data. And so reverse all other presented tables revise and uniform when possible.

Answer: All tables in uniform alignment are presented

- Specify the sections for histopathological examination in brain tissue , add the score of different detected lesions.

Answer: Histopathological analysis is revised and pictures at higher magnification 40X are incorporated. 

Reviewer #2: Authors revised the manuscript partially, still few improvement required.

1. Suggested to perform one-way ANOVA, authors stated that they have performed. But the information such as F- value, which post-hoc analysis carried out is missing.

Answer: We performed one-way ANOVA and add the level of significance as p value as in our previous published articles. In next work we will also add F-value. Now it is difficult to again analyze whole manuscript results and add F-value. Kindly consider for this time. 

2. The Table-2, reporting the change in the protein level between the experimental group. This analysis is very basic, measuring the total protein content is not considered as research data. This table should be removed.

Answer: Table 2 is removed. 

3. Figure 2 & 3, the immunohistochemistry data is not clear, provided image is only 10X. At least, to see some details the magnification should be 40X or 60X.

Answer: Histopathological images at 40X magnification are incorporated.

---

## [Decision Letter · Decision Letter 1]

3 Jan 2024

PONE-D-23-32989R1Biochemical properties and biological potential of Syzygium heyneanum with Antiparkinson’s activity in paraquat induced rodent modelPLOS ONE

Dear Dr. Chauhdary,

Thank you for submitting your manuscript to PLOS ONE. After careful consideration, we feel that it has merit but does not fully meet PLOS ONE’s publication criteria as it currently stands. Therefore, we invite you to submit a revised version of the manuscript that addresses the points raised during the review process. Please submit your revised manuscript by Feb 17 2024 11:59PM. If you will need more time than this to complete your revisions, please reply to this message or contact the journal office at plosone@plos.org. Please include the following items when submitting your revised manuscript:A rebuttal letter that responds to each point raised by the academic editor and reviewer(s). You should upload this letter as a separate file labeled 'Response to Reviewers'.A marked-up copy of your manuscript that highlights changes made to the original version. You should upload this as a separate file labeled 'Revised Manuscript with Track Changes'.An unmarked version of your revised paper without tracked changes. You should upload this as a separate file labeled 'Manuscript'.If applicable, we recommend that you deposit your laboratory protocols in protocols.io to enhance the reproducibility of your results. Protocols.io assigns your protocol its own identifier (DOI) so that it can be cited independently in the future. For instructions see: https://journals.plos.org/plosone/s/submission-guidelines#loc-laboratory-protocols. Additionally, PLOS ONE offers an option for publishing peer-reviewed Lab Protocol articles, which describe protocols hosted on protocols.io. Read more information on sharing protocols at https://plos.org/protocols?utm_medium=editorial-email&utm_source=authorletters&utm_campaign=protocols.

We look forward to receiving your revised manuscript.

Kind regards,

Yasmina Abd‐Elhakim

Academic Editor

PLOS ONE

Journal Requirements:

Reviewers' comments:

Reviewer's Responses to Questions

**Comments to the Author**

1. If the authors have adequately addressed your comments raised in a previous round of review and you feel that this manuscript is now acceptable for publication, you may indicate that here to bypass the “Comments to the Author” section, enter your conflict of interest statement in the “Confidential to Editor” section, and submit your "Accept" recommendation.

Reviewer #1: (No Response)

Reviewer #2: All comments have been addressed

2. Is the manuscript technically sound, and do the data support the conclusions?

Reviewer #1: (No Response)

Reviewer #2: Partly

3. Has the statistical analysis been performed appropriately and rigorously? 

Reviewer #1: (No Response)

Reviewer #2: No

4. Have the authors made all data underlying the findings in their manuscript fully available?

Reviewer #1: (No Response)

Reviewer #2: Yes

5. Is the manuscript presented in an intelligible fashion and written in standard English?

Reviewer #1: (No Response)

Reviewer #2: Yes

6. Review Comments to the Author

**Reviewer #1:** The authors addressed most of the required points, while still some major changes are required :

- The Table format should be unique in the manuscript , the tables presented are not the same.

- The table footnote are also not unique.

- The rationale of the selected doses should be rewritten in the methods section.

- Table (7) : it is not suitable to add a single column in the table. Also table 8 , and 9 , all should be merged in one table

- Figure (2) : is not well presented and not separated well , the photos in the plate should be organized as done in Figure (3)

**Reviewer #2:** Authors have mentioned that they have performed "One-way or two-way ANOVA, followed by the Tukey's post-test for multiple comparisons ". However, the F- value and post hoc analysis analysis data not reported in the manuscript. I am wondering why the authors not stored their analyzed data. i) They have to remove the statement from method section, the statement will mislead the readers. ii) They have to report in the manuscript, the limitation of the study is need to validate with "One-way or two-way ANOVA, followed by the post-hoc test".

7. PLOS authors have the option to publish the peer review history of their article (what does this mean?). If published, this will include your full peer review and any attached files.

Reviewer #1: No

Reviewer #2: No

---

## [Author Response · Author response to Decision Letter 1]

23 Jan 2024

Dear reviwer 2, we add F- value in statistical analysis, as you suggested, kindly consider it now

---

## [Editor Report · Decision Letter 2]

2 Feb 2024

Biochemical properties and biological potential of Syzygium heyneanum with Antiparkinson’s activity in paraquat induced rodent model

PONE-D-23-32989R2

Dear Dr. Chauhdary,

We’re pleased to inform you that your manuscript has been judged scientifically suitable for publication and will be formally accepted for publication once it meets all outstanding technical requirements.

Kind regards,

Yasmina Abd‐Elhakim

Academic Editor

PLOS ONE
---

## [Editor Report · Acceptance letter]

20 Feb 2024

PONE-D-23-32989R2 

PLOS ONE

Dear Dr. Chauhdary, 

I'm pleased to inform you that your manuscript has been deemed suitable for publication in PLOS ONE. Congratulations! Your manuscript is now being handed over to our production team.

Kind regards, 

on behalf of

Prof. Dr. Yasmina Abd‐Elhakim 

Academic Editor

PLOS ONE